# Inhibitors of Lipoxygenase and Cyclooxygenase-2 Attenuate Trimethyltin-Induced Neurotoxicity through Regulating Oxidative Stress and Pro-Inflammatory Cytokines in Human Neuroblastoma SH-SY5Y Cells

**DOI:** 10.3390/brainsci11091116

**Published:** 2021-08-24

**Authors:** Woo-Ju Song, Jang-Hyuk Yun, Myeong-Seon Jeong, Kil-Nam Kim, Taekyun Shin, Hyoung-Chun Kim, Myung-Bok Wie

**Affiliations:** 1College of Veterinary Medicine and Institute of Veterinary Science, Kangwon National University, Chuncheon 24341, Korea; songwooju@kangwon.ac.kr (W.-J.S.); yunjh@kangwon.ac.kr (J.-H.Y.); 2Chuncheon Center, Korea Basic Science Institute, Chuncheon 24341, Korea; jms0727@kbsi.re.kr (M.-S.J.); knkim@kbsi.re.kr (K.-N.K.); 3Department of Biochemistry, Kangwon National University, Chuncheon 24341, Korea; 4Department of Veterinary Anatomy, College of Veterinary Medicine and Veterinary Medical Research Institute, Jeju National University, Jeju 63243, Korea; shint@jejunu.ac.kr; 5Neuropsycopharmacology and Toxicology Program, College of Pharmacy, Kangwon National University, Chuncheon 24341, Korea; kimhc@kangwon.ac.kr

**Keywords:** apoptosis, cyclooxygenase-2, esculetin, lipoxygenase, trimethyltin

## Abstract

Trimethyltin (TMT) is an environmental neurotoxin that mediates dopaminergic neuronal injury in the brain. In this study, we characterized the toxic mechanism and possible protective compounds against TMT-induced neurotoxicity in human dopaminergic neuroblastoma SH-SY5Y cells. Antioxidants such as melatonin, N-acetylcysteine (NAC), α-tocopherol, and allopurinol alleviated TMT toxicity. Apoptosis induced by TMT was identified by altered expression of cleaved caspase-3, Bax, Bcl-2, and Bcl-xL through Western blot analysis. The iron chelator deferoxamine ameliorated the alteration of apoptosis-related proteins through TMT exposure. TMT also induced delayed ultrastructural necrotic features such as mitochondrial swelling and cytoplasmic membrane rupture; NAC reduced these necrotic injuries. Esculetin, meloxicam, celecoxib, and phenidone decreased TMT toxicity. Elevation of the pro-inflammatory cytokines IL-1β, TNF-α, and NF-ĸB and reduction of the antioxidant enzymes catalase and glutathione peroxidase-1 (GPx-1) were induced by TMT and ameliorated by inhibitors of LOX and COX-2 enzymes. Both NMDA and non-NMDA antagonists attenuated TMT toxicity. The free calcium ion modulators nimodipine and BAPTA/AM contributed to neuronal survival against TMT toxicity. Inhibitors of the phosphoinositide 3-kinase/protein kinase B/mammalian target of rapamycin pathway, an autophagy regulator, decreased TMT toxicity. These results imply that TMT neurotoxicity is the chief participant in LOX- and COX-2-mediated apoptosis, partly via necrosis and autophagy in SH-SY5Y cells.

## 1. Introduction

Trimethyltin (TMT) represents a group of organotin compounds that are commonly used as biocides or stabilizers in agricultural and industrial fields [1,2]. Accordingly, TMT is considered a risk factor for food safety and may cause toxicological issues in humans and ecosystems [3]. The accumulation of TMT in the brain has been known as correlated with delayed excitability in the central nervous system, and the behavioral abnormalities derived from TMT intoxication mainly exhibit tremor and convulsion [3]. However, despite the execution of various challenges on the mechanistic investigation of TMT, the exact etiological causes and preventive strategies are not established until now. Although TMT induces neuronal damage in the various brain regions such as the hippocampus, piriform/entorhinal cortex, amygdala, olfactory bulb, and pyramidal cells of the neocortex in animals, only a few studies have explored the mechanism of TMT-induced toxicity or survival strategies of dopaminergic neurons in these regions. Mignini et al. [4] reported that TMT decreased dopamine (DA) receptors and DA transporters in the hippocampus, followed by cognitive dysfunction. TMT has also been reported to decrease DA turnover in the caudate nucleus, a portion of the basal ganglia [5], as well as DA concentration in the nucleus accumbens [6]. Notably, dopaminergic neurons exhibit selective vulnerability via a range of oxidative stress factors [7]; therefore, neurotoxicity studies of TMT using the human dopaminergic neuroblastoma SH-SY5Y cell line are needed to establish suitable preventive strategies for neurodegenerative diseases. The neuropathological symptoms evoked by TMT are also associated with neurobehavioral abnormalities, such as seizures, hyperactivity, deficits in learning and memory [8], and aggression. TMT can increase intracellular calcium levels via excitotoxicity in spiral ganglion cells; nifedipine, an L-type calcium channel antagonist, decreases intracellular calcium accumulation [9]. TMT leads to the breakdown of homeostasis of intracellular calcium concentration via internal stores, such as the mitochondria and endoplasmic reticulum (ER) in human neuroblastoma SH-SY5Y cells [10]. Excitatory glutamate receptors such as N-methyl-D-aspartate (NMDA) and α-amino-3-hydroxy-5-methyl-4-isoxazolepropionic acid/kainate (AMPA/KA) receptors are distributed in SH-SY5Y cells [11,12,13]. TMT treatment has been shown to result in neuronal necrosis; the anti-inflammatory agent dexamethasone failed to inhibit neuronal damage in the mouse hippocampus [14]. However, autophagy and caspase-dependent apoptosis occur in hippocampal regions as a result of TMT-induced neuronal injury [15]. Autophagic vacuoles were present in neurons after TMT exposure [16], against which the autophagy activators rapamycin and lithium protected neuronal cells. Glutamate receptor-mediated excitotoxicity may be an inducing factor for necrosis through intracellular calcium overload in SH-SY5Y neuroblastoma cells [12]. Gunasekar et al. [17] reported that TMT induced apoptosis or necrosis in a dose-dependent manner in cerebellar granule cells. Concurrently, they suggested that various oxidative stresses including protein kinase C (PKC) activation, overproduction of nitric oxide (NO) and hydrogen peroxide, and overstimulation of metabotropic glutamate receptors may be involved in necrotic death induced by TMT exposure. Previously, we reported that TMT selectively increased protein kinase C delta (PKCδ) expression through various oxidative stresses in the hippocampus [18]. TUNEL-positive apoptosis and various oxidative injuries, such as increases in malodialdehyde (MDA), protein carbonyl, and reactive oxygen species (ROS), were found in the hippocampal area. Apoptosis induced by TMT depends on the balance between NF-ĸB and MAP kinases in SH-SY5Y cells [19]. Accordingly, we explored the types of cell death after TMT exposure in human dopaminergic neuroblastoma SH-SY5Y cells. In this study, we characterized the dose-toxicity effects of TMT using lactate dehydrogenase (LDH), MTT, and MDA assays; the alteration of protein expression on pro-apoptosis and antiapoptosis factors such as cleaved caspase-3, Bax, Bcl-2, and Bcl-xL; ultrastructural cell changes; changes in pro-inflammatory cytokines such as IL-1β, TNF-α, and NF-ĸB and antioxidant enzymes such as catalase and GPx-1; potential neuroprotectants such as antioxidants and inhibitors of lipoxygenase (LOX) and/or cyclooxygenase-2 (COX-2), glutamate receptor blockers, free calcium ion modulators, iron chelators, pan-caspase inhibitors, and PI3K/Akt/mTOR autophagy signaling pathway inhibitors; ultrastructural changes in transmission electron microscopy (TEM) morphology induced by TMT in human dopaminergic neuroblastoma SH-SY5Y cells.

## 2. Materials and Methods

### 2.1. Reagents, Chemicals, and Antibodies

Trimethyltin chloride (TMT) was purchased from Thermo Fisher Scientific (Ward Hill, MA, USA). Fetal bovine serum (FBS), antibiotic–antimycotic (100X), and Dulbecco’s modified Eagle medium (DMEM) were purchased from Gibco Co. (Grand Island, NY, USA). Esculetin, meloxicam, α-tocopherol, melatonin, N-acetylcysteine, allopurinol, deferoxamine, Z-VAD-fmk, CNQX (6-cyano-7-nitorqunoxalline-2,3-dione), dextrorphan tartrate, BAPTA/AM (1,2-Bis(2-aminophenoxy)ethane-N,N,N’,N’-tetraacetic acid tetrakis(acetoxymethyl ester)), MTT [3-(4,5-dimethylthiazol-2-yl)-2,5-diphenyltetrazolium bromide], and Triton X-100 were purchased from Sigma Chemical Co. (St. Louis, MO, USA). Celecoxib and rapamycin were purchased from Cayman Co. (Ann Arbor, MI, USA) and Enzo Life Sciences (Farmingdale, NY, USA), respectively. Primary anti-cleaved caspase-3, anti-Bcl-2, anti-Bcl-xL, and anti-Bax antibodies were obtained from Cell Signaling Technology (Danvers, MA, USA). Anti-β-tubulin and peroxidase-linked secondary antibodies were obtained from Santa Cruz Biotechnology (Dallas, TX, USA).

### 2.2. Cell Culture and TMT Treatment

The human neuroblastoma SH-SY5Y cell line was purchased from the Korean Cell Line Bank (KCLB) of Seoul National University (Seoul, Korea) and cultured in DMEM containing 10% FBS, as well as an antibiotic–antimycotic solution containing 100 μg/mL streptomycin, 100 U/mL penicillin, and 0.25 μg/mL Fungizone. The cells were incubated in a humidified atmosphere with 5% CO_2_ at 37 °C. The cells were harvested using 0.25% trypsin EDTA and were sub-cultured into 100 mm culture dishes. Grown cells were seeded into 100 mm dishes and 24- and 96-well plates. TMT was dissolved in dimethyl sulfoxide (DMSO). Each control was added to the same volume of DMSO as a vehicle-treated group. Cells were exposed to TMT for 12 h for TEM observation and for 24 h for other analyses. Esculetin, meloxicam, celecoxib, and phenidone were pre-incubated for 2 h prior to TMT treatment and maintained in co-exposure with TMT and test chemicals for 24 h. Other test chemicals were co-exposed with TMT for 24 h. Toxicity was evaluated at the highest concentration of each test chemical. 

### 2.3. Photomicrography of Cultured Cells

Representative photomicrographs in vehicle control treated or 1, 20, or 30 μM TMT-treated cells were taken using an inverted microscope (Olympus, Tokyo, Japan).

### 2.4. Lactate Dehydrogenase (LDH) Activity Measurement

Cells were seeded in 24-well plates at a cell density of 2.5 × 10^5^ cells/well. After 1 day, the growth media were changed to DMEM without FBS. To determine the moderate toxic concentration of TMT, cells were treated with 0.3, 1, 3, 5, 10, 20, 30, 50, 100, or 150 μM of TMT for 24 h. Cell injury was quantitatively estimated based on LDH release from damaged cells into the culture medium. The toxicity of each chemical was evaluated using LDH assays. LDH activity was measured at a wavelength of 340 nm using a kinetic program on a VersaMax Eliza Reader (Molecular Devices, San Jose, CA, USA).

### 2.5. MTT Assay

Cells were seeded in a 96-well culture plate at a density of 1 × 10^4^ cells/well and cultured in DMEM with 10% FBS and antimycin at 37 °C in a 5% CO_2_ incubator. After 2 days, cells were transferred to DMEM without FBS and exposed to TMT and/or rescuing chemicals in a 200 µL volume and then incubated for 20 h. Next, the MTT solution was added and the cells were incubated at 37 °C for 3 h. After discarding the culture medium, 100 µL of dimethyl sulfoxide (DMSO) was added and the cells were incubated for 30 min to dissolve formazan crystals. The viability of the cells was estimated at 570 nm using a VersaMax microplate reader (Molecular Devices). 

### 2.6. Western Blot Analysis

Treated cells at a density of 3 × 10^6^ cells/100 mm dish containing TMT alone or co-exposed to TMT and deferoxamine were harvested, lysed in a lysis buffer, and subjected to sodium dodecyl sulfate–polyacrylamide gel electrophoresis (SDS–PAGE). The blots were incubated with the designated primary antibodies. Horseradish peroxidase-conjugated species-specific IgGs were used as secondary antibodies. The blots were incubated with an enhanced chemiluminescence substrate (Thermo Fisher Scientific) and exposed to film.

### 2.7. Enzyme-Linked Immunoassay (ELISA)

To measure the activities of IL-1β, TNF-α, NF-ĸB SOD, and GPx-1, a human enzyme-linked immunoassay (ELISA) assay kit (CUSABIO, Hubei Province, China) was used according to the manufacturer’s instructions. Briefly, IL-1β and TNF-α activities were assessed by collecting 100 µL of media from SH-SY5Y cells. To assess the activities of NF-ĸB, SOD, and GPx-1, SH-SY5Y cells were treated with Pierce^TM^ RIPA Buffer (Thermo Fisher Scientific, Waltham, MA, USA) and Halt^TM^ Protease Inhibitor Cocktail (Thermo Fisher Scientific, Waltham, MA, USA), and sonicated for 3 min ON and 30-s OFF cycles. The cycle was repeated three times. After sonication, the cells were centrifuged at 12,000 rpm for 20 min at 4 °C, and 100 µL of supernatant was transferred to a new tube. Next, 100 µL of media (IL-1β, TNF-α) or supernatant (NF-ĸB, SOD, GPx-1) and standard were added to specific-antibody-coated 96-well microplates and incubated at 37 °C in a 5% CO_2_ incubator for 2 h. All supernatant was removed, 100 µL of 1× biotin antibody was added to each well, and the plates were incubated at 37 °C in a 5% CO_2_ incubator for 1 h. Each well was then washed three times with 1× wash buffer, 100 µL 1× horseradish peroxidase–avidin was added, and the plates were incubated at 37 °C in a 5% CO_2_ incubator for 1 h. Each well was washed five times, 90 µL of TMB (3,3′,5,5′-tetramethylbenzidine) substrate was added, and the plates were incubated at 37 °C in a 5% CO_2_ incubator for 25 min in the dark. Following the addition of 50 µL of stop solution, the activities of IL-1β, TNF-α, NF-ĸB, SOD, and GPx-1 were estimated at 450 nm using a VersaMax microplate reader (Molecular Devices). 

### 2.8. Protein Assay

To measure the amount of protein from SH-SY5Y cells, a bicinchoninic acid (BCA) Protein Assay Kit TAKARA BIO, Inc., Nojihigashi, Japan) was used according to the manufacturer’s instructions. The supernatant was obtained from sonicated and centrifuged cells, and 100 µL of supernatant, standard, and working solution were added to a 96-well microplate and incubated at 60 °C for 1 h. After 1 h, the amount of protein was measured at 562 nm using a VersaMax microplate reader (Molecular Devices).

### 2.9. TEM Observations

Prior to TEM observations, the cultured cells were washed with 0.1 M phosphate-buffered saline (PBS) and fixed with a mixture of 1% paraformaldehyde and 4% glutaraldehyde overnight at 4 °C. Cells were post-fixed in 1% osmium tetroxide in the same buffer and dehydrated with ethanol and propylene oxide. Subsequently, the samples were embedded in Epon-812 resin and ultra-thin sections were obtained using an ultra-cut microtome (Leica Co., Greenwood Village, CO, USA). Finally, sections were stained with uranyl acetate and lead citrate and subjected to TEM visualization (LEO912AB, Carl Zeiss, Oberkochen, Germany). 

### 2.10. Statistical Analyses

All statistical analyses were conducted using the SAS 9.4 program. Statistical analyses consisted of one-way analysis of variance (ANOVA) tests and Tukey’s multiple comparison test, at a significance level of *p* < 0.05. All experimental data are expressed as means ± standard error of the mean (SEM). All experiments were performed at least three times with similar results.

## 3. Results

### 3.1. Dose-Toxicity of Trimethyltin (TMT)

TMT cytotoxicity assayed using MTT (Figure 1A) and LDH (Figure 1B) displayed dose-dependent decreases and increases, respectively, at concentrations between 0.3 and 150 µM. Malondialdehyde (MDA) showed significant dose-dependent increases at concentrations of TMT greater than 3 µM (Figure 1C). Significant suppression of quantitative viability was observed at TMT concentrations of 1–3 µM of TMT. We selected 10 µM of TMT as an experimental control for rescue studies with test chemicals based on moderate LDH release following TMT treatment. 

### 3.2. Photomicrographs of Representative Cell Morphologies

Representative cellular images induced by TMT treatment are shown in Figure 2A–D. There were no morphological changes at 1 µM TMT (Figure 2B), compared to the control (Figure 2A). Morphologically, delayed loss or injury to neurites and cell body swelling were distinct at 10 µM TMT (Figure 2C). Most cell bodies were shrunken and lysed at 30 µM TMT (Figure 2D). The median toxic dose (TD_50_) was quantified based on LDH activity at a concentration of 10 µM. Therefore, we selected 10 µM as the experimental control concentration for all neuroprotective studies.

### 3.3. Melatonin, N-acetylcysteine (NAC), α-tocopherol, and Allopurinol Protected against TMT-Induced Neurotoxicity in SH-SY5Y Cells

Melatonin, a pineal gland hormone in the brain, has protective effects at concentrations between 100 and 200 µM (Figure 3A). NAC (a glutathione precursor) and α-tocopherol attenuated TMT toxicity at concentrations of 0.3–2 mM and 50–300 µM, respectively (Figure 3B,C). Allopurinol, a hydroxyl radical scavenger, also ameliorated TMT-induced neurotoxicity at 300 µM (Figure 3D).

### 3.4. Deferoxamine Reversed Altered Expression of Cleaved Caspase-3, Bax, Bcl-2, and Bcl-xL Induced by TMT

Western blot analysis revealed that the expression of cleaved caspase-3, Bcl-2, Bcl-xL, and Bax proteins in SH-SY5Y cells was altered by TMT treatment with or without deferoxamine. The exposure of cells to TMT significantly upregulated the pro-apoptotic proteins cleaved caspase-3 and Bax and downregulated the antiapoptotic proteins Bcl-2 and Bcl-xL (Figure 4A,B). Among these proteins, cleaved caspase-3 showed greater alteration. The iron chelator deferoxamine significantly recovered to the control state upon altered expression of apoptosis-related proteins. 

### 3.5. NAC Improved TMT-Exposed SH-SY5Y Cells Observed via TEM

Ultrastructural observations of SH-SY5Y cells in the control group showed generally healthy organelles with intact mitochondria and cytosolic membranes (Figure 5A,B). TMT induced typical necrotic damage with mitochondrial swelling (Figure 5D) and cytosolic membrane rupture (Figure 5C) at 12 h after exposure. Co-exposure of cells to TMT and NAC showed undamaged mitochondria and the appearance of autophagolysosomes (Figure 5F).

### 3.6. Esculetin, Meloxicam, Celecoxib, and Phenidone Attenuated TMT-Induced Neurotoxicity in SH-SY5Y Cells

A selective lipoxygenase (LOX) inhibitor, esculetin, showed a significant and dose-dependent protective effect on TMT-induced neurotoxicity (Figure 6A). Meloxicam and celecoxib, selective COX-2 inhibitors, also had significant protective effects against TMT toxicity (Figure 5B,C). Phenidone, an inhibitor of both LOX and COX-2, exhibited remarkable inhibition against TMT neurotoxicity (Figure 6D).

### 3.7. Esculetin and Meloxicam Inhibited the Elevation of Pro-inflammatory Cytokines IL-1β, TNF-α, and NF-kB Induced by TMT in SH-SY5Y Cells

TMT induced a concentration-dependent increase in the pro-inflammatory cytokines IL-1β, TNF-α, and NF-kB (Figure 7A,D,G). Esculetin, a LOX inhibitor (Figure 7B,E,H), and meloxicam, a COX-2 inhibitor (Figure 7C,F,I), significantly ameliorated the increase in the pro-inflammatory cytokines IL-1β, TNF-α, and NF-kB induced by TMT.

### 3.8. Esculetin and Meloxicam Ameliorated Catalase (CAT) and Glutathione Peroxidase-1 (GPx-1) Activity Reduction in SH-SY5Y Cells

TMT dose-dependently decreased the activity of antioxidant enzymes CAT and GPx-1 in SH-SY5Y cells (Figure 8A,D). Esculetin (10–30 μM) and meloxicam (50 μM) significantly recovered the reductions in CAT (Figure 8B,C) and GPx-1 (Figure 8E,F). However, 300 μM meloxicam failed to reverse the reduction in GPx-1 activity.

### 3.9. Dextrorphan and CNQX Attenuated TMT-Induced Neurotoxicity in SH-SY5Y Cells

CNQX, a competitive AMPA/KA receptor antagonist, showed a significant neuroprotective effect against TMT toxicity at concentrations of 30–100 μM (Figure 9A). Dextrorphan, a non-competitive NMDA receptor antagonist, also had a protective effect against TMT-induced toxicity at a concentration of 50 μM (Figure 9B). 

### 3.10. Calcium and Iron Chelators and A Voltage-gated L-type Calcium Channel (VGCC) Blocker Attenuated TMT Toxicity in SH-SY5Y Cells

The cell-permeable intracellular calcium chelators [1,2-Bis(2-aminophenoxy)ethane-N,N,N’,N’-tetraacetic acid tetrakis(acetoxymethyl ester)] and BAPTA/AM (Figure 10A), and an iron chelator, deferoxamine (Figure 10B), showed protective effects against TMT-induced toxicity in SH-SY5Y cells. The voltage-gated L-type calcium channel blocker nimodipine (10–30 μM) also significantly ameliorated TMT-induced neurotoxicity (Figure 10C). 

### 3.11. Inhibitors of Pan-Caspase, PI3K, Akt, and mTOR Attenuated TMT-Induced Toxicity in SH-SY5Y Cells

The pan-caspase inhibitor Z-Vad-fmk (Figure 11A), phosphatidylinositol 3-kinase (PI3K) inhibitor LY294002 (Figure 11B), Akt translocation inhibitor BML257 (Figure 11C), and mTOR inhibitor rapamycin (Figure 11D) significantly inhibited TMT-induced neurotoxicity. A hypothesis based on these results is illustrated in Figure 12. 

## 4. Discussion

In this study, we discovered that ROS production and subsequent neuroinflammation induced by TMT may be correlated with the eicosanoid pathways via both LOX and COX-2 enzymes. The inhibition of LOX by esculetin, COX-2 by meloxicam, and both LOX and COX-2 by phenidone may represent an important strategy for the maintenance of neuronal viability from TMT-induced brain injury. TMT decreased the expression of dopamine receptors D1 and D2, as well as dopamine transporters, inducing impairment of spatial reference memory in rat hippocampal areas [4]. Thus, most dopaminergic neurons in the hippocampus may be target sites vulnerable to exposure to TMT derived from various industrial organotin compounds. In apoptotic injury, the cleaved caspase-3 protein exhibited eightfold overexpression due to TMT exposure, the highest overexpression among all apoptosis regulators including Bax, Bcl-2, and Bcl-xL proteins. Therefore, cleaved caspase-3 apoptosis protein is the most sensitive factor in TMT toxicity among SH-SY5Y cells. These results suggest that caspase may be a more excellent parameter than LDH. In antiapoptosis proteins, decreased expression of Bcl-2 and Bcl-xL, which play important roles in inhibiting mitochondria-dependent apoptosis, was observed following TMT treatment. The iron chelator deferoxamine inhibited these alterations of apoptosis and antiapoptosis proteins evoked by TMT, which suggests that the production of hydroxyl free radicals and MDA via Fenton reactions may be closely involved in TMT-induced apoptosis. In these results, there is necessary further study to clarify the exact profiles of the molecular mechanism of TMT neurotoxicity using quantitative real-time polymerase chain reaction (qPCR) assay. In a previous study, melatonin was found to ameliorate TMT-mediated neuroinflammation in vivo [20]. The antioxidant ascorbate improved TMT-induced seizures by regulating glutathione homeostasis and various oxidative stresses including MDA [21]. Similarly, we confirmed that melatonin, a singlet oxygen radical scavenger, and NAC, a glutathione precursor, as well as the radical scavengers α-tocopherol and allopurinol, inhibited TMT-induced neuronal injury. In contrast to these results, TMT has been reported to trigger necrosis and autophagy based on ultrastructural observations in vivo [22]. This necrosis was chiefly localized to neurons but not glial or endothelial cells. In microstructural TEM studies using SH-SY5Y cells, we demonstrated that neuronal cell injury morphology induced by TMT was characterized by delayed typical necrosis, indicated by mitochondrial swelling and cytoplasmic membrane rupture.

These microstructural necrotic findings induced by TMT improved with exogenous glutathione supplements such as NAC. Notably, autophagolysosomes were found in the group co-treated with TMT and NAC. These results suggest that the pattern of TMT-induced neuronal cell death mainly shows apoptosis, with partial necrosis and autophagy components in human dopaminergic neuroblastoma SH-SY5Y cells. TMT-induced swelling of SH-SY5Y cells was observed in phase-contrast images. In other in vivo studies, TMT was found to induce extensive necrosis, such as extensive neuronal edema, lysosome accumulation, and myelinoid membranous bodies at pyramidal neurons in the neonatal rat hippocampus [23]. Some studies have reported increases in COX-2 expression in the CA1 region of the rat hippocampus [24,25]. Houng et al. [26] also reported that indomethacin, a COX inhibitor, alleviated TMT-induced neuronal injury in the dentate gyrus of mice. In the present study, we demonstrated, for the first time, that the LOX inhibitor esculetin, as well as COX-2 inhibitors and an inhibitor of both LOX and COX-2—namely, meloxicam, celecoxib, and phenidone—protected SH-SY5Y cells from TMT-induced neurotoxicity. TMT increases the expression of pro-inflammatory cytokines such as interleukin-1β (IL-1β), tumor necrosis factor-α (TNF-α), and nuclear factor-ĸB (NF-ĸB) in neurons, glia, and microglia after TMT-induced hippocampal injury [19,27,28,29,30,31,32]. In these studies, IL-1β, TNF-α, and NF-ĸB dose-dependently increased after TMT exposure. Pre-treatment with meloxicam or esculetin significantly reduced the increase in pro-inflammatory cytokines after TMT exposure. Notably, after TMT exposure, meloxicam and esculetin each potently reversed the elevation of IL-1β and TNF-α to the control state. In epilepsy models, meloxicam diminished IL-1β and TNF-α levels [33]. Esculetin and meloxicam showed similar inhibitory tendencies toward the pro-inflammatory transcription factor, NF-ĸB. TMT reduced the antioxidant enzyme activities of SOD, CAT, and GPx, including GSH levels, in the rat brain [34,35]. CAT and GPx-1 activity induced by TMT exposure decreased dose-dependently in our culture system. Esculetin and meloxicam showed significant reversal effects. Glutamate receptors, including NMDA and non-NMDA receptors, participate in excitotoxic injury in SH-SY5Y cells [12]. Neuronal injury induced by TMT treatment was ameliorated by antagonists of non-NMDA and NMDA receptors. These results suggest that excitotoxic neuronal injury may partly contribute to TMT-induced cell death. However, we observed that 0.5 mM NMDA, 0.3 mM KA, and 5 mM glutamate did not induce neuronal injury in SH-SY5Y cells (Appendix A). These results suggest that the distribution and function of glutamate receptors in SH-SY5Y cells are insufficient to induce excitotoxic injury and are consistent with those of other studies that have reported excitotoxicity at glutamate concentrations of > 20 mM [12]. TMT also stimulated calcium-mediated glutamate release in brain-slice cultures, while nifedipine, an L-type calcium channel blocker, and MK-801, a non-competitive NMDA channel blocker, did not relieve the glutamate efflux [36]. However, we observed that another L-type calcium channel blocker, nimodipine, and an intracellular calcium chelator, BAPTA-AM, significantly ameliorated TMT-induced neurotoxicity. This discrepancy may partly be due to the difference between the high concentrations (0.1–1 mM) of TMT used by Dawson et al. [36] and the 10 µM concentration used in the present study. Recently, Rakshit et al. [37] reported that deferoxamine, an iron-chelator, protects neuronal cells from 6-hydroxydopamine-induced apoptosis and autophagy in SH-SY5Y cells. In the present study, deferoxamine inhibited TMT-induced neurotoxicity and apoptosis, whereas the pan-caspase inhibitor z-VAD-fmk suppressed TMT-induced neuronal injury. Signaling of the PI3K/Akt/mTOR pathway plays an important role in neuronal survival or neurodegeneration via autophagy or apoptosis processes [38,39]. In human neuroblastoma cells, relatively high activation of the PI3K/Akt/mTOR signaling pathway has been reported [40]. In this study, we showed that rapamycin, an mTOR inhibitor and autophagy activator, contributes to neuronal survival against TMT-induced neurotoxicity. By contrast, we observed that autophagic LC3 intensity did not change within 24 h after 10 μM TMT treatment, which differed from the pharmacological results (data not shown). Autophagy inhibitors severely exacerbated TMT-induced neurotoxicity in neuronal cell cultures, in contrast to our results [16]. In another study, a potent autophagy inducer, rapamycin, protected PC12 cells through mTOR inhibition [41], indicating that autophagy contributes to the increased viability of cells during neuronal damage. These experiments have been executed by only one cell line. Accordingly, further comparative studies through other cell lines, such as animal or human neuronal cell lines will be necessary in the near future. Conclusively, these results demonstrate that TMT-induced neurotoxicity is involved in LOX- and COX-2-mediated apoptosis, and may participate in necrosis or autophagy via calcium-mediated oxidative stress and pro-inflammatory cytokines.

## 5. Conclusions

In this study, TMT showed caspase-dependent apoptosis and calcium-dependent necrosis. Some antioxidants, LOX and COX-2 inhibitors, chelators of iron and calcium, calcium channel blockers, NMDA and AMPA/KA receptor blockers, and PI3K/Akt/mTOR pathway modulators could be potential candidate compounds for therapy of TMT intoxication. In particular, inhibitors of LOX and COX-2 exhibited neuroprotective effects via regulating oxidative stress and pro-inflammatory cytokines in human dopaminergic neuroblastoma SH-SY5Y cells.

## Figures and Tables

**Figure 1 brainsci-11-01116-f001:**
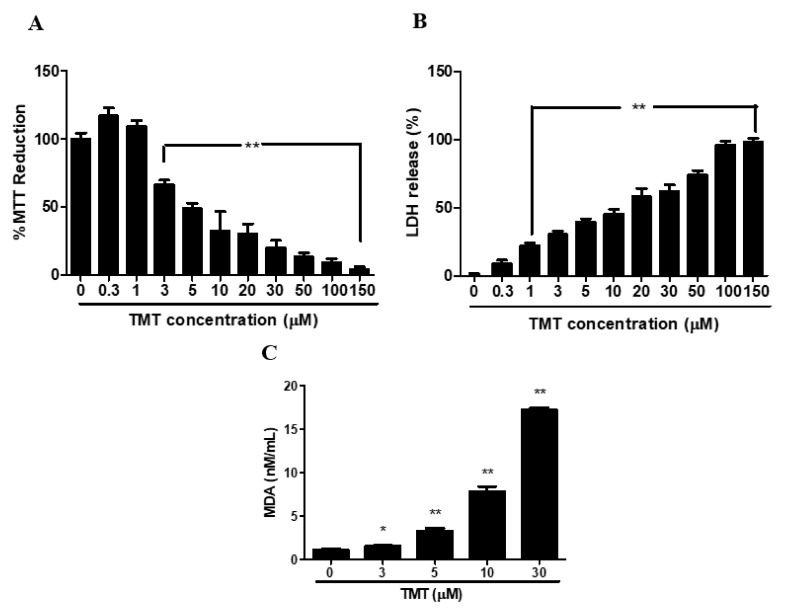
Dose-toxicity effects of trimethyltin (TMT) in human dopaminergic neuroblastoma SH-SY5Y cells. Cell viability and lipid peroxidation induced by TMT were measured by MTT reduction (**A**), lactate dehydrogenase (LDH) (**B**), and malondialdehyde (MDA) assays (**C**). * *p* < 0.05, ** *p* < 0.01 compared to control group (*n* = 4). Experiments were performed at least three times, with similar results. n: number of wells in each group.

**Figure 2 brainsci-11-01116-f002:**
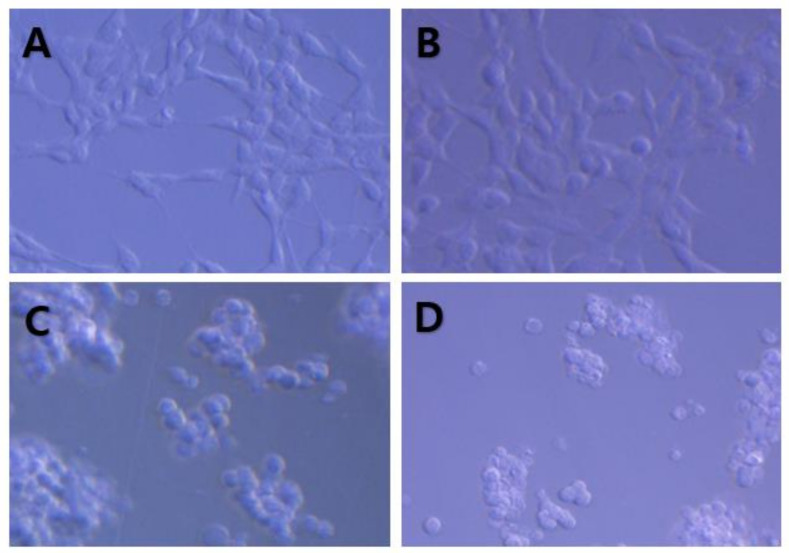
Representative morphological phase-contrast photomicrographs of vehicle- (control) (**A**) or TMT-exposed cells (**B**–**D**). TMT was administered as vehicle control (**A**), or at concentrations of 1 μM (**B**), 10 μM (**C**), or 30 μM (**D**). No or weak neuronal injuries were observed at a TMT concentration of 1 μM, whereas most neurons were massively damaged, with delayed swelling, neurite and cell body loss, and lysis at TMT concentrations of 10 and 30 μM.

**Figure 3 brainsci-11-01116-f003:**
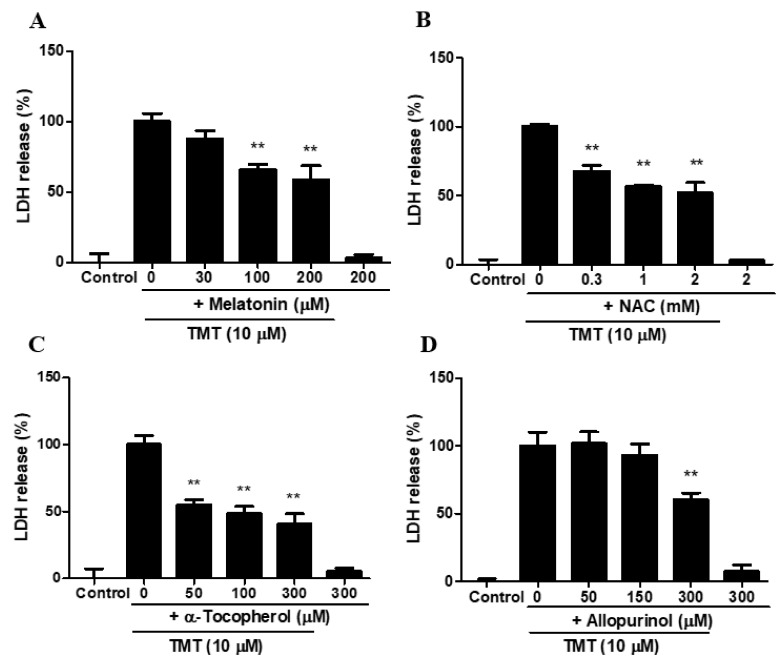
Protective effects of various antioxidants melatonin (**A**), N-acetylcysteine (NAC) (**B**), α-tocopherol (**C**), and allopurinol (**D**) against TMT-induced neurotoxicity in SH-SY5Y cells. Cytotoxicity was evaluated using a lactate dehydrogenase (LDH) release assay. ** *p* < 0.01 compared to the 10-μM TMT-treated group (*n* = 4). Experiments were performed at least three times, with similar results.

**Figure 4 brainsci-11-01116-f004:**
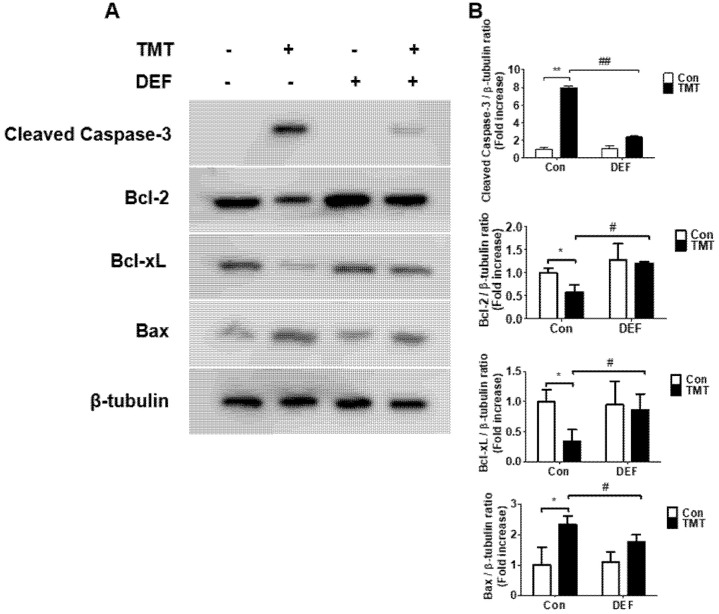
(**A**) Alterations of cleaved caspase-3, Bcl-2, Bcl-xL, and Bax levels induced by 10 μM TMT were determined by Western blot analysis. β-tubulin was used as a control. Deferoxamine showed significant inhibition of TMT-induced alteration of apoptosis and antiapoptosis proteins. (**B**) Histogram showing quantitative densitometric analysis. * *p* < 0.05 and ** *p* < 0.01 (*n* = 3), compared to vehicle (DMSO)-treated group. # *p* < 0.05 and ## *p* < 0.01 (*n* = 3), compared to TMT-treated group. Experiments were performed at least three times, with similar results. Con: control, DEF: deferoxamine, DMSO: dimethyl sulfoxide.

**Figure 5 brainsci-11-01116-f005:**
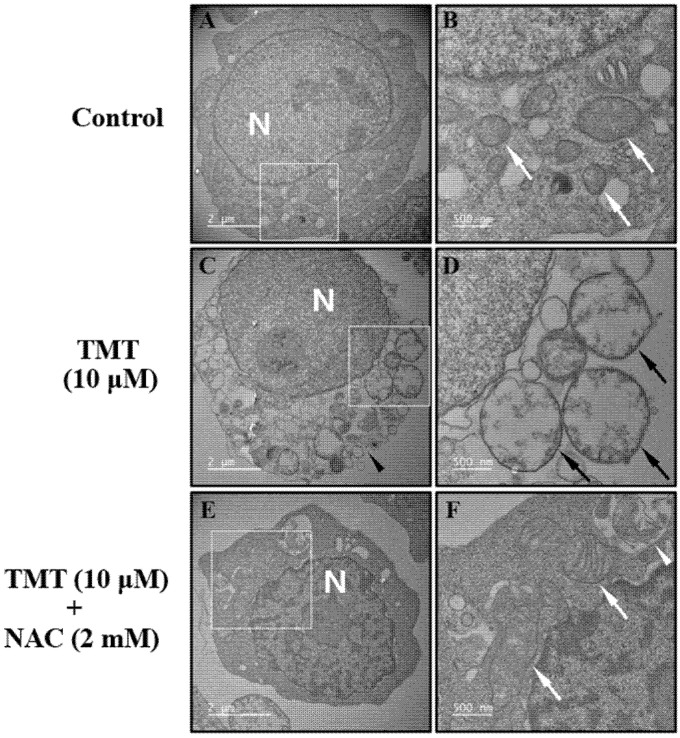
Transmission electron microscopy (TEM) images of SH-SY5Y cells treated with vehicle as control (**A**,**B**), 10-μM TMT (**C**,**D**), and 10-μM TMT + 2 mM NAC (**E**,**F**). (**B**,**D**,**F**) are magnified white-square portions of (**A**,**C**,**E**), respectively. The white arrow and pointer in (**B**,**F**) indicate normal mitochondria and autophagolysosome, respectively. The black arrow and pointer in (**C**,**D**) indicate swollen mitochondria and ruptured cytoplasmic membrane. N: nucleus.

**Figure 6 brainsci-11-01116-f006:**
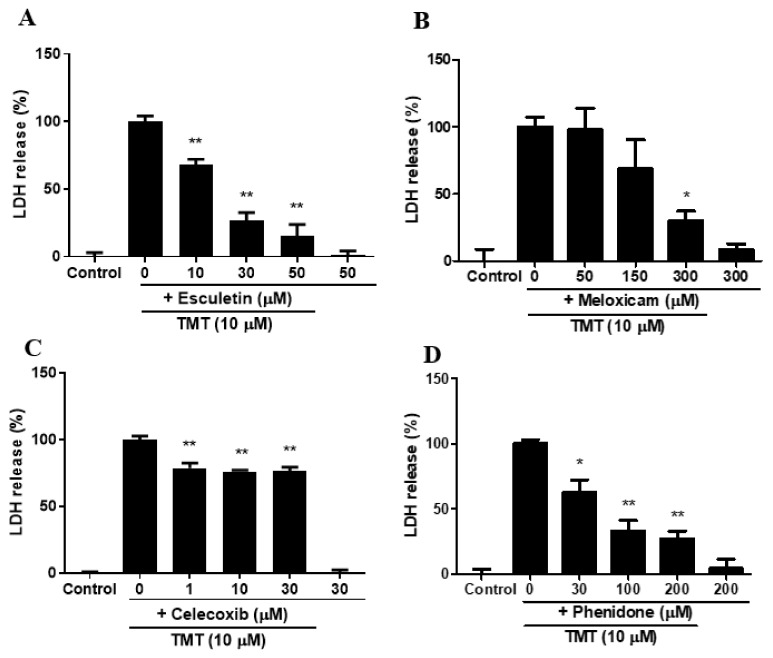
Lipoxygenase (LOX) inhibitor, esculetin (**A**); cyclooxygenase-2 (COX-2) inhibitors, meloxicam (**B**) and celecoxib (**C**); and inhibitor of both LOX and COX-2, phenidone (**D**), significantly attenuated TMT-induced neurotoxicity in SH-SY5Y cells. * *p* < 0.05 and ** *p* < 0.01 compared to 10-μM TMT-treated group (*n* = 4). Experiments were performed at least three times, with similar results.

**Figure 7 brainsci-11-01116-f007:**
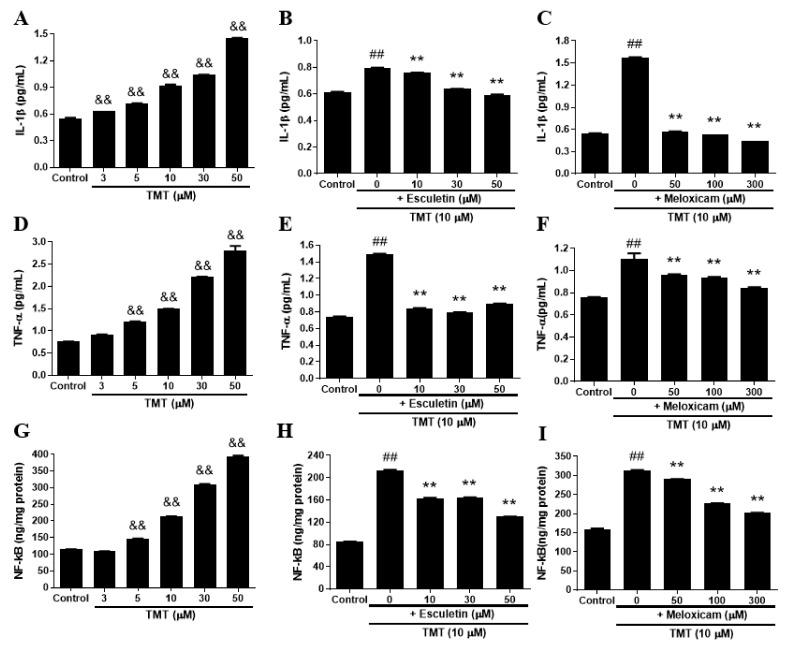
TMT induced a concentration-dependent increase in pro-inflammatory cytokines, interleukin-1β (IL-1β) (**A**), tumor necrosis factor (TNF-α) (**D**), and NF-kB (**G**) in SH-SY5Y cells. Esculetin (**B**,**E**,**H**) and meloxicam (**C**,**F**,**I**) significantly inhibited TMT-induced elevation of IL-1β, TNF-α, and NF-kB, respectively. ^&&^
*p* < 0.01 compared to vehicle group (*n* = 4). ** *p* < 0.01 compared to 10 µM TMT-treated group (*n* = 4). ^##^
*p* < 0.01 compared to control group (*n* = 4). Experiments were performed at least three times, with similar results.

**Figure 8 brainsci-11-01116-f008:**
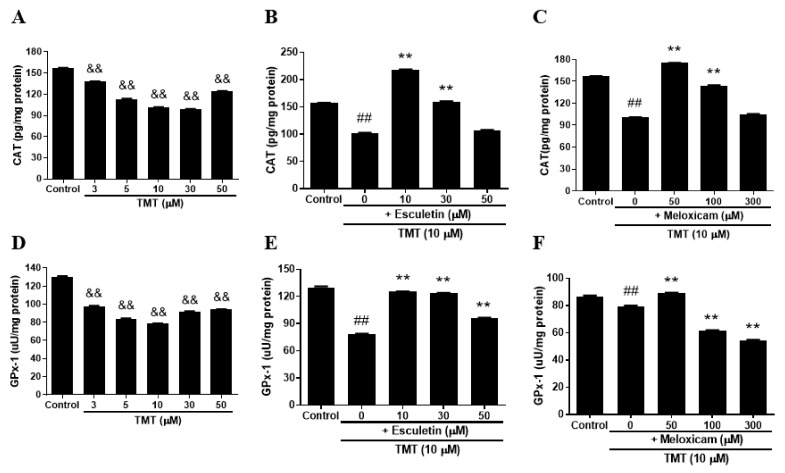
TMT decreased endogenous antioxidant enzymes, catalase (CAT) (**A**), and glutathione peroxidase-1 (GPx-1) (**D**) in SH-SY5Y cells. Esculetin (**B**,**E**) and meloxicam (**C**,**F**) significantly reversed TMT-mediated reduction of CAT and GPx-1. ^&&^
*p* < 0.01 compared to the vehicle group (*n* = 4). ** *p* < 0.01 compared to the 10 µM TMT-treated group (*n* = 4). ^##^
*p* < 0.01 compared to the control group (*n* = 4). Experiments were performed at least three times, with similar results.

**Figure 9 brainsci-11-01116-f009:**
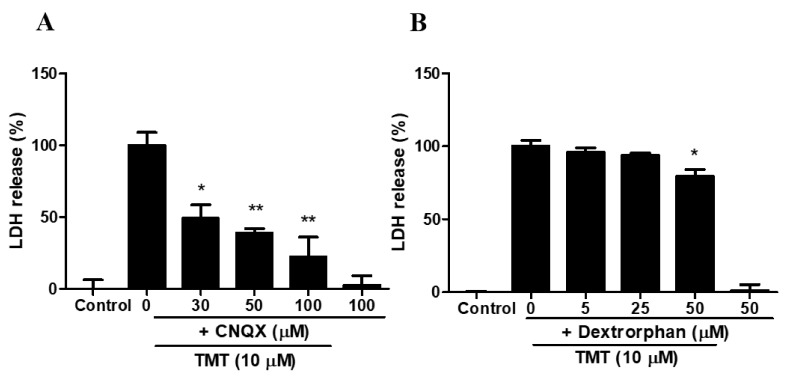
CNQX, an AMPA/KA receptor antagonist (**A**) and dextrorphan, a non-competitive NMDA antagonist (**B**), attenuated TMT-induced neuronal damage in SH-SY5Y cells. Cytotoxicity was evaluated using a lactate dehydrogenase (LDH) release assay. * *p* < 0.05 and ** *p* < 0.01 compared to the 10 µM TMT-treated group (*n* = 4). Experiments were performed at least three times, with similar results.

**Figure 10 brainsci-11-01116-f010:**
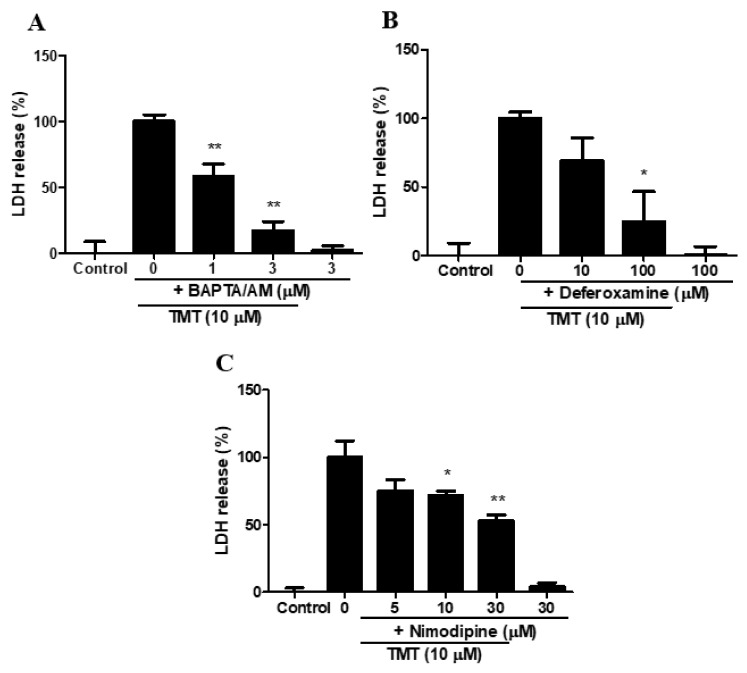
Protective effects of BAPTA/AM (**A**), deferoxamine (**B**), and nimodipine (**C**), on TMT-induced neurotoxicity in SH-SY5Y cells. Cytotoxicity was evaluated using a lactate dehydrogenase (LDH) release assay. * *p* < 0.05, ** *p* < 0.01 compared to 10 µM TMT-treated group (*n* = 4). Experiments were performed at least three times, with similar results.

**Figure 11 brainsci-11-01116-f011:**
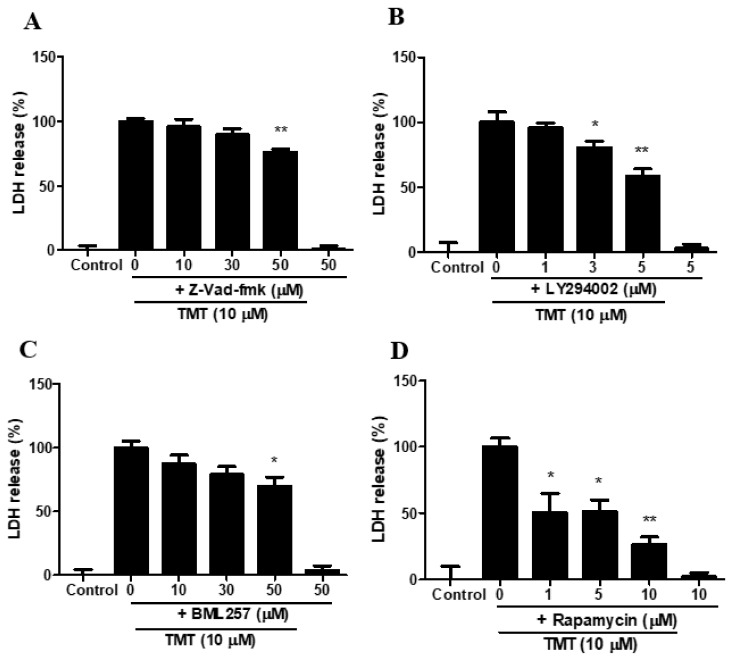
Protective effects of Z-Vad-fmk (**A**), LY294002 (**B**), BML257 (**C**), and rapamycin (**D**) against TMT-induced neurotoxicity in SH-SY5Y cells. Cytotoxicity was evaluated using a lactate dehydrogenase (LDH) release assay. * *p* < 0.05, ** *p* < 0.01 compared to 10 µM TMT-treated group (*n* = 4). Experiments were performed at least three times, with similar results.

**Figure 12 brainsci-11-01116-f012:**
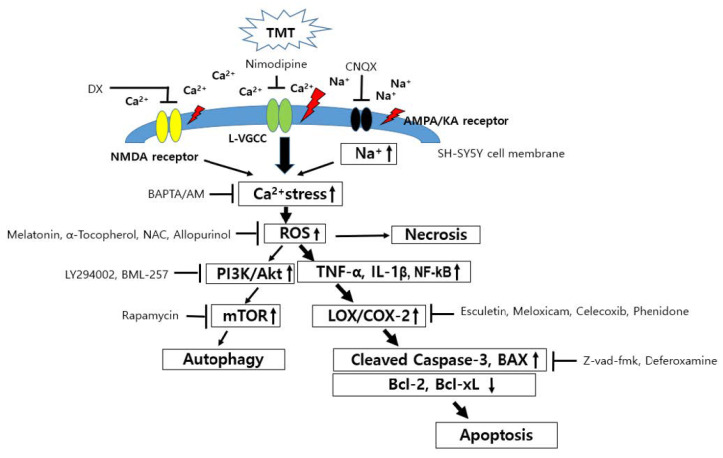
Hypothesis of a TMT-induced neurotoxic mechanism in human neuroblastoma SH-SY5Y cells.

## Data Availability

Data is contained within the article or Appendix A.

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
