# Peer review of "Inhibitors of Lipoxygenase and Cyclooxygenase-2 Attenuate Trimethyltin-Induced Neurotoxicity through Regulating Oxidative Stress and Pro-Inflammatory Cytokines in Human Neuroblastoma SH-SY5Y Cells"

_brainsci, 2021, doi:10.3390/brainsci11091116_

Round 1

Reviewer 1 Report

Manuscript ID: brainsci-1326867

Trimethyltin Exhibits Lipoxygenase- and Cyclooxygenase-2-mediated Apoptosis via Oxidative Stress and Pro-inflammatory Cytokines in Human Neuroblastoma SH-SY5Y Cells

RECOMMENDATION – MAJOR REVISION

Disadvantages of the paper:

  • The title is misleading. The Authors verifying here the capacity of several compound to reduce the toxicity of TMT
  • The aim of the study is not new. There are several papers indicating neurotoxic capacity of TMT in the context of apoptosis, neuroinflammation etc.
  • The novelty is the drug verification. The Authors should pay more attention to this part of the paper.
  • The introduction should be more informative.

Comments:

  • The resolution of the figures is very low. Please correct this.
  • Please add the whole western blots photos as a supplement.
  • 4 part B first graph – is there only * between TMT groups?
  • 4 – there is no * in the description of the figure
  • Please provide the information what the “n” means and how many replicates the Authors used.

Author Response

Response to the referee’s comments

Thank you for your kind review and valuable comments on our manuscript, which we have carefully revised in response to your comments and suggestions.

Reviewer(s)’ Comments to Author:

[Reviewer 1]

Manuscript Ref. No.: brainsci-1326867

Manuscript title: Trimethyltin Exhibits Lipoxygenase- and Cyclooxygenase-2-mediated Apoptosis via Oxidative Stress and Pro-inflammatory Cytokines in Human Neuroblastoma SH-SY5Y Cells

  1. The title is misleading. The Authors verifying here the capacity of several compounds to reduce the toxicity of TMT.

Response: We totally agree with your sharp criticism. The goal of this study chiefly focused to find the optimal protectants against TMT-mediated neuronal injuries in SH-SY5Y cells. It was difficult to decide a suitable title from the results of several neuroprotective compounds as you indicated. Accordingly, we selected the key results obtained from inhibitors of representative inflammatory enzymes, such as cyclooxygenase-2 (COX-2) and lipoxygenase (LOX) against TMT-induced neuronal damage as a title of this study. So, we cautiously changed the title of this manuscript as “Inhibitors of Lipoxygenase and Cyclooxygenase-2 Attenuate Trimethyltin-Induced Neurotoxicity through Regulating Oxidative Stress and Pro-inflammatory Cytokines in Human Neuroblastoma SH-SY5Y Cells” as reviewer indicated. 

  1. The aim of this study is not new. There are several papers indicating neurotoxic capacity of TMT in the context of apoptosis.

Response: As reviewer’s indication, there were lots of apoptotic results in TMT neurotoxicity.

         However, examination on changes of apoptosis-related proteins, such as cleaved

  caspase-3, Bcl-2, Bcl-xL and Bax in human dopaminergic neuroblastoma SH-SY5Y cells is deficient so far. Moreover, we newly observed that ultrastructural findings by TMT exposure were necrosis as on the contrary to apoptotic changes. Comparison among inhibitor of LOX and COX-2, and both inhibitor of LOX and COX-2 did not execute or published until now. Moreover, these results suggest that excitotoxic insults may partly involve in TMT toxicity although the distribution of glutamate receptors in dopaminergic SH-SY5Y cells might be scarce. We have already reported that co-localization of apoptosis and necrosis may be occur in delayed-type neuronal death by restriction of intracellular free calcium.

  1. The novelty is the drug verification. The Authors should pay more attention

to this part of the paper.

Response: We agree with your opinion on our paper. Accordingly, we changed the title in line

with drug verification as reviewer’s comments.  

  1. The introduction should be more informative.

Response: According to reviewer’s comment, we inserted additional informative contents in

introduction part.

Comments

  1. The resolution of the figures is very low. Please correct this.

Response: According to reviewer’s comment, we changed phase-contrast photomicrographs

 (Fig. 2B, 2C, and 2D) into more cleared photos. All bar graph from Fig. 1 to Fig.

 11 was also changed as more cleared blackened bar graph.

  1. Please add the whole western blots photos as a supplement.

Response: According to reviewer’s comment, we newly added the whole western blots photos

 as a supplemental Fig. S4.

  1. 4 part B first graph-is there only * between TMT groups?

Response: According to reviewer’s comment, we checked statistical analysis data and changed the significance marking, “*” and “#” into “**” and “##”, respectively at first cleaved caspase-3 graph. The marking of significance between TMT itself group and TMT + DEF group also changed “*” into “#” in residual B figures.

  1. 4-there is no * in the description of the figure.

Response: According to reviewer’s comment, we inserted the explanation of “*” marking

significance description in the legend of Fig. 4.

  1. Please provide the information what the “n” means and how many replicates the Authors used.

Response: According to reviewer’s comment, we added “n” as “number of wells” in first Fig. 1. This abbreviation like “n” is usually use without special explanation in most cellular neuroscience journals. All experiments were performed at least 3 times, with similar results. These phrases already inscribed at the legends of figures.

Reviewer 2 Report

Paper needs major revision.

Over one cell line is needed to prove the drug effect.

Reporter assays are needed to see transcriptional activation of molecular targets that authors claimed

Please use vehicle DMSO as control.

Author Response

Response to the referee’s comments

Thank you for your kind review and valuable comments on our manuscript, which we have carefully revised in response to your comments and suggestions.

Reviewer(s)’ Comments to Author:

[Reviewer 2]

  1. Over on cell line is needed to prove the drug effect.

Response: Your indication is very correct. According to reviewer’s comment, hippocampal cell line, such as HT22 could be one of the best choice. However, neurotoxicity study already executed many times by other researchers. So, we have scheduled to examine the TMT-induced vulnerability and availability of its possible neuroprotectants on dopaminergic neurons which exhibits sensitivity on oxidative stress in hippocampal areas.

  1. Reporter assays are needed to see transcriptional activation of molecular targets that authors claimed.

  Response: We totally agree with reviewer’s sharp indication on the verification of molecular target to demonstrate the molecular mechanism of TMT neurotoxicity and its inhibitors. However, we could not carry out these Luciferase Reporter assays due to shortage of technical researcher and research fund.

  1. Please use vehicle DMSO as control.

 Response: We always added same volume of vehicle DMSO in every experiments. In this study, there is no difference between vehicle and control. According to reviewer’s comment, we added these contents at 2.2. Cell culture and TMT treatment in Materials and Methods.

Round 2

Reviewer 1 Report

The Authors answered all my questions/objections.

Author Response

Thank you for your valuable comment on our manuscript, which we carefully send an answer sheet in response to reviewer’s comments and suggestions.

Reviewer(s)’ Comments to Author:

[Reviewer 1]

Manuscript Ref. No.: brainsci-1326867

Comments: The Authors answers all my questions/objections

Response: We deeply appreciate on the positive comments and generous criticism.

Reviewer 2 Report

Authors did good job in revising the manuscript. I suggest to apply qPCR method to assess targeted genes expression. Caspase assay is more sensitive in estimation of  cell death than LDH one.

I did not find an explanation for why one cell line is applied. 

Author Response

Thank you for your second valuable comment on our manuscript, which we carefully send an answer sheet in response to reviewer’s comments and suggestions.

Reviewer(s)’ Comments to Author:

[Reviewer 2]

  1. Comment 1: Author did a good job in revising the manuscript. I suggest to apply qPCR method to assess targeted genes expression

Response: We appreciate on your generous criticism. We partly agree with your suggestion about new data production from qPCR assays for verification of targeted gene expression. As you estimated in title, the goal of our studies chiefly focus on the comparison of pharmacological efficiency among potential neuroprotective compounds instead of mechanism study on TMT-induced neurotoxicity. It is difficult to carry on qPCR assays due to end of research grant and transfer of lab member. Thus, we added these limitations at discussion part at page 14, line 4-6. Please politely ask your generosity on our confined situation in view of this point.

  1. Comment 2: Caspase assay is more sensitive in estimation of cell death than LDH one.

Response: I agree with your comment. However, LDH assay is also useful viability measuring tools for excitotoxicity- or necrosis-mediated neuronal injury in case of saturation of caspase enzymes during neuronal injury. In our previous studies, caspase activity was increased according to escalate zinc oxide nanoparticle-induced neuronal injury state. Here, it was impossible to compare with caspase activity among damaged groups in later stage of neuronal injury (reached plateau state very fast). Accordingly, caspase also appears limitation depends on the type of neuronal injury. Here, we knew that caspase may be a sensitive tool in TMT-induced neurotoxic state. In later injury stage, LDH was more stable parameter than caspase one. Furthermore, apoptosis also may include caspase-dependent and caspase-independent component. In our caspase assay using Western blot analysis, TMT increased about 8-times in caspase activity. These results suggest caspase assay seems very sensitive factor among other apoptosis-relating proteins, as reviewer indicated. Thus, we added reviewer’s indication and concern in discussion part at page 13, line 21-22.

  1. Comment 3: I did not find an explanation for why one cell line is applied.

  Response: We already have answered about this at No. 1 response.  According to reviewer’s comment, hippocampal cell line, such as HT22 could be one of the best choice. However, neurotoxicity study already executed several times by other researchers. Zhang et al. (2006) have reported TMT-relating paper using HT22 hippocampal cell line. We usually interested in the field of Parkinson’s disease. So, we have scheduled to examine the TMT-induced vulnerability and availability of its possible neuroprotectants on the vulnerability of dopaminergic neurons which exhibits sensitivity on oxidative stress in hippocampal areas. Thus, we added your indication at discussion part at page 17, line 4-6.

This manuscript is a resubmission of an earlier submission. The following is a list of the peer review reports and author responses from that submission.

Round 1

Reviewer 1 Report

The manuscript entitled Trimethyltin induces apoptosis and necrosis via oxidative stress and pro-inflammatory cytokines in SH-SY5Y neuroblastoma cells illustrates extensive approach to reveal mechanisms TMT acts in cells. Trimethyltin chloride (TMT) occurs in polyvinyl chloride and silicone products, such as kitchen utensils, food packages, and fungicides.  Authors stated they studied the dose-toxicity effects of TMT; changes in pro-inflammatory cytokines, such as IL-1β, TNF-α, and NF-ĸB and antioxidant enzymes, such as catalase and GPx-1; useful neuroprotectants; coexistence of apoptosis, autophagy, and necrosis using pharmacological assay measures; and changes in TEM morphology to prevent neuronal injury induced by TMT in human dopaminergic neuroblastoma SH-SY5Y cells. The authors explored the types of cell death after TMT exposure in human dopaminergic neuroblastoma SH-SY5Y cells. Is it specific only to SH-SY5Y cells? I recommend using at least two other neuroblastoma models to prove it. How authors prove necrosis in cells?

Second, I did not find the length of TMT treatment. Does it affect cytotoxicity? Also, please note antioxidants were preincubated with cells or added together with TMT?

Third, please note the methods you used to distinguish apoptosis from necrosis. I recommend including flow cytometry analysis of cells treated TMT and simple western blots to see caspases-2, -3 -6, Bax, Bcl-2, and other core proteins involved in apoptosis.

Reviewer 2 Report

Comments:

  1. The country is not added to the affiliations.
  2. How the TMT concentration used by the Authors (10 uM TMT) coincides with human exposure to the TMT?
  3. How much of TMT is released into the environment (e.g. per year)?
  4. Does the cell line has been checked against mycoplasm?
  5. The n=4 is too small for biochemical analyses, especially since the 3R procedure does not limit the Authors (thanks to the use of cell lines).
  6. The resolution of the figures is low. The figures are blurred.
  7. The neuroprotective compounds are used in very high concentrations. Why?
  8. There is no data how the neuroprotective compounds alone act in SH-SY5Y.
  9. The information quote “ after 2 or 3 days, TMT or other compounds were added…” is too general. Please, specify this.